# Potential Role of Sphingolipidoses-Associated Lysosphingolipids in Cancer

**DOI:** 10.3390/cancers14194858

**Published:** 2022-10-05

**Authors:** Patricia Dubot, Leonardo Astudillo, Nicole Therville, Lorry Carrié, Magali Pettazzoni, David Cheillan, Jérôme Stirnemann, Thierry Levade, Nathalie Andrieu-Abadie, Frédérique Sabourdy

**Affiliations:** 1Université de Toulouse, Inserm, CNRS, Université Toulouse III-Paul Sabatier, Centre de Recherches en Cancérologie de Toulouse, Equipe Labellisée Fondation ARC, 2 Avenue Hubert Curien, CS 53717, CEDEX 1, 31037 Toulouse, France; 2Laboratoire de Biochimie Métabolique, CHU Toulouse, 31059 Toulouse, France; 3Service de Médecine Interne, Clinique Saint Exupéry, 31077 Toulouse, France; 4Laboratoire de Biochimie, Hospices Civils de Lyon, 69000 Bron, France; 5Division of General Internal Medicine, Department of Medicine, Geneva University Hospitals, 1211 Geneva, Switzerland

**Keywords:** cancer, sphingolipid, glucosylsphingosine, Gaucher disease, psychosine, lysoGb3, lysosulfatide, melanoma, sphingosylphosphocholine

## Abstract

**Simple Summary:**

Sphingolipidoses are a subgroup of rare inherited disorders of lipid metabolism, most often due to a lysosomal enzymatic defect affecting sphingolipid catabolism. They are characterized by the accumulation of sphingolipids and their deacylated derivatives, called lysosphingolipids. An increased occurrence of cancer is suspected in some of these disorders. Whether lysosphingolipids are involved is unknown, and there is little data available on this topic. Here, we review the literature on lysosphingolipids and their potential effects on cellular properties, which could contribute to cancer development. Finally, we present data showing that one of them (namely, glucosylsphingosine) induces modifications of melanoma cells, which may favor tumor progression. Our review highlights that a better knowledge of lysosphingolipids, the biomarkers of rare diseases, could contribute to a better understanding of the pathophysiology of cancer.

**Abstract:**

Sphingolipids play a key structural role in cellular membranes and/or act as signaling molecules. Inherited defects of their catabolism lead to lysosomal storage diseases called sphingolipidoses. Although progress has been made toward a better understanding of their pathophysiology, several issues still remain unsolved. In particular, whether lysosphingolipids, the deacylated form of sphingolipids, both of which accumulate in these diseases, are simple biomarkers or play an instrumental role is unclear. In the meanwhile, evidence has been provided for a high risk of developing malignancies in patients affected with Gaucher disease, the most common sphingolipidosis. This article aims at analyzing the potential involvement of lysosphingolipids in cancer. Knowledge about lysosphingolipids in the context of lysosomal storage diseases is summarized. Available data on the nature and prevalence of cancers in patients affected with sphingolipidoses are also reviewed. Then, studies investigating the biological effects of lysosphingolipids toward pro or antitumor pathways are discussed. Finally, original findings exploring the role of glucosylsphingosine in the development of melanoma are presented. While this lysosphingolipid may behave like a protumorigenic agent, further investigations in appropriate models are needed to elucidate the role of these peculiar lipids, not only in sphingolipidoses but also in malignant diseases in general.

## 1. Introduction

Sphingolipids (SLs) comprise a very large number of amphiphilic lipid molecules, having in common a long chain aminoalcohol referred to as a sphingoid base. They are present in plants and animals, particularly in mammals. Owing to their chemical structure, most of them are constituents of cell membranes and preferentially locate in the outer leaflet of the plasma membrane. Similar to diacylglycerolipids, some of the major SLs are also components of circulating lipoproteins. Of note, sphingosine 1-phosphate (S1P) is typically found in the extracellular compartment, being relatively abundant in plasma and lymph. Membrane SLs, which are, by far, the predominant SL species, contain one molecule of fatty acid. In accordance with previous classifications of lipids, where the term “lyso” denotes a glycero(phospho)lipid lacking a radyl group [1], lysosphingolipids (lysoSLs) are defined as SLs that lack the fatty acyl moiety.

SLs are synthesized by all mammalian cells. Their biosynthesis starts in the endoplasmic reticulum (ER) by the condensation of an acyl-CoA, mainly palmitoyl-CoA, and L-serine to form the backbone of the sphingoid base. Reduction of this initial product leads to sphinganine, which is N-acylated, i.e., linked to a fatty acyl moiety through an amide bond by a ceramide synthase. After desaturation, the resulting ceramide molecule can be further transformed into complex SLs, such as sphingomyelin or glycosphingolipids. Whereas sphingomyelin and β-glucosylceramide-based glycolipids are synthesized in the Golgi apparatus, β-galactosylceramide is formed in the ER. They all are transported to the plasma membrane. The degradation of diet-derived SLs is mediated by secreted intestinal enzymes. As to cellular SLs, their physiological catabolism mostly occurs in the lysosomes through a conserved sequence of hydrolytic steps (see Figure 1). In this pathway, the breakdown of complex SLs gradually releases the residues of the hydrophilic headgroup attached to the ceramide backbone. The last step of lysosomal degradation is catalyzed by acid ceramidase, which cleaves the amide bond and, thus, liberates sphingosine and fatty acids.

Knowledge of SLs and SL metabolism has been strongly stimulated by the existence in humans of genetic conditions characterized by the accumulation of selective undegraded SL molecules. Extensive biochemical and genetic studies on these diseases, named sphingolipid storage disorders (or sphingolipidoses), led to the identification of their underlying defects more than 50 years ago. These disorders result from the disruption of the lysosomal catabolic pathway due to the deficient function of one of the hydrolases, either because this enzyme or its so-called activator protein (sphingolipid activator protein) is mutated, leading to profoundly decreased catalytic activity [2,3]. The catabolic pathway implicating these enzymatic and activator proteins is depicted in Figure 1, and the lipid molecules that accumulate in the corresponding sphingolipid storage disorders are listed in Table 1. As the entry of SLs into the endolysosomal compartment is permanent, although variable, depending on cell type and external conditions, the enzymatic defect translates into substrate accumulation in the lysosomes (see Table 1). Abnormally elevated concentrations of the undegraded SLs can be observed in organs as well as in biological fluids such as blood and urine; the presence of the accumulated SLs in the extracellular milieu may be explained by cell damage or lysosomal exocytosis.

The clinical presentation of patients affected with sphingolipid storage disorders is quite diverse, ranging from isolated visceral symptomatology to severe and fatal neuronopathic forms. A factor that determines organ involvement is the tissue distribution of the accumulated SLs. For instance, a neurological disease develops when the metabolism of GM1 and GM2 gangliosides or sulfatides and galactosylceramide, which are key components of neurons and myelin, respectively, are not properly degraded. On the other hand, the defective turnover of glucosylceramide (GlcCer) or sphingomyelin (SM), which are widely distributed, affects cells such as professional phagocytes, i.e., monocytes-macrophages. Of importance for the discussion below, the age of disease onset can vary widely, ranging from the antenatal period up to late adulthood. This variability is explained, at least partially, by the residual activity of the affected enzyme: the lower the activity is, the more severe the lipid storage and the symptomatology are [4].

As already mentioned, the deficient activity of a lysosomal hydrolase results in the accumulation of its primary substrate. However, biochemical studies performed as early as in the mid-seventies have also documented the presence and storage of deacylated SLs, e.g., galactosylsphingosine (GalSph), also called psychosine, and glucosylsphingosine (GlcSph), in organs of patients affected with Krabbe and Gaucher diseases, respectively [5,6]. The discovery of these two lysoSLs was soon followed by the observation that other similarly N-deacylated SLs accumulate in distinct sphingolipidoses (see Table 1). Abnormally high levels of lysoSLs are also found in patients’ biological fluids such as plasma, urine, and cerebrospinal fluid. The metabolic source of lysoSLs has long been debated. Initially, it was postulated that the synthesis of a lysoSL follows the same biosynthetic route as the conventional lipid but uses a sphingoid base rather than ceramide. Recent evidence has shown that instead, GalSph, GlcSph and lysoGb3 are generated by the action of acid ceramidase on their corresponding SL in the lysosomal compartment, i.e., where the parental lipid is stored [7,8]. In fact, the elimination of GalSph production by ablating acid ceramidase in a mouse model of Krabbe suppressed the behavioral and histopathological features of leukodystrophy [9]. Whether other lysoSLs are produced by acid ceramidase still requires further investigation. The fatty acid amide hydrolase (FAAH) enzyme has been reported to partially account for the production of lysosulfatide [10].

Finally, for the purpose of the present paper, one should recall that sphingosine 1-phosphate (S1P) is also a lysoSL. This lysophospholipid is mainly formed by the sphingosine-kinase-mediated phosphorylation of sphingosine and by the autotaxin-mediated breakdown of lysoSM [11]. It is still unknown whether a ceramidase can generate S1P from ceramide 1-phosphate. So far, with the possible exception of Gaucher disease, neither accumulation of S1P nor ceramide 1-phosphate has been reported in sphingolipid storage disorders.

As some sphingolipid storage diseases are associated with the occurrence of malignancies and because the lysolipid S1P is well known to behave as a second messenger, mediating survival and pro-angiogenic pathways and being involved in tumorigenesis, one can ask the question of the potential role of lysoSLs in cancer. The following chapters will discuss the biological properties of lysoSLs related to cancer and their potential contribution to the development of malignancies that are associated with some sphingolipid storage diseases.

## 2. Cancer Prevalence in Sphingolipid Storage Disorders

Gaucher disease (GD) is the most frequent sphingolipid storage disease, of which the deficiency of lysosomal glucosylceramidase (GCase), encoded by the *GBA* gene, results in an accumulation of GlcCer and GlcSph (Table 1). Three disease forms are described: type 1 is the most frequent, with hepatosplenomegaly, anemia, thrombocytopenia, bone abnormalities, and lung complications without neurological involvement, the latter being only present in types 2 and 3 [28] (Table 2). Numerous case reports, case series, and cohort studies have described the occurrence of malignancies in Gaucher patients and have suggested that the risk of developing cancer is increased in Gaucher patients with a relative risk evaluated at 1.7 (95% confidence interval (95% CI) 1.3–2.3) [29]. In the French Gaucher Disease Registry (FGDR) (n = 657 patients), we report that 27 patients presented malignancies (4.1%) [30]. Compared to data from the French population (obtained from the International Agency for Research on Cancer (https://gco.iarc.fr/today/home) accessed on 31 March 2021), our observations suggest a marked increase in the prevalence of malignancies at an odds ratio (OR) of 1.8 (95% CI 1.03–2.81) (Table 3). The risk of multiple myeloma seems to be particularly marked, with an OR of 23.8 (95% CI 4.9–70.2) (see Table 2 and Table 3), which is very consistent with the literature [29,31,32,33]. Nevertheless, these results should be interpreted with caution because of limits relative to study design. In addition, an association with monoclonal gammopathy of undetermined significance (MGUS), a pre-malignant condition for multiple myeloma, is also suspected [32]. In the FGDR, MGUS has been found in 59 patients (25%) [34]. Besides the risk of hematological malignancies, the prevalence of digestive cancer and, especially, hepatocellular carcinoma in Gaucher patients also seems to be increased [35]. However, in the FGDR, no hepatocellular carcinoma has been observed. Interestingly, in the FGDR, five Gaucher patients presented two cancers, a finding that has been previously described [31,33,36] (see Table 3).

All these observations would suggest that GD could be a predisposing condition for the development of cancers.

The risk of cancer could also be increased in patients suffering from other sphingolipidoses. Indeed, some case reports [37,38,39,40,41,42,43] and cohort studies [44,45] have also described the development of cancer in Fabry patients. Fabry disease (FD), the second most frequent sphingolipid storage disease, is an X-linked disorder resulting from the deficient activity of lysosomal alpha-galactosidase A, encoded by the *GLA* gene, leading to an accumulation of Gb3 and lysoGb3 (Table 1). FD is classically characterized by angiokeratoma, cornea verticillata, neuropathy (acroparesthesia), progression toward chronic kidney disease, cerebrovascular accidents, and hypertrophic cardiomyopathy (Table 2). In British (n = 261) [44] and Italian (n = 53) [46] cohorts of Fabry patients, an increased incidence was observed for urological and kidney cancers, as well as melanoma, compared to the general population [44]. An increase in the occurrence of meningioma, a benign tumor, has also been observed (see Table 2) [44,45].

The occurrence of cancer in Niemann–Pick disease types A and B (NPD-A and B) could also be non-negligible. NPD-A and -B are very rare diseases that result from the deficient activity of acid sphingomyelinase (ASM), encoded by the *SMPD1* gene, leading to an accumulation of SM and lysoSM (Table 1). Clinical phenotypes range from a very severe phenotype (NPD-A), with neurovisceral involvement and an early death, to a mild phenotype (NPD-B), with hepatosplenomegaly, cytopenia, and lung involvement. Very recently, an increase in the prevalence of cancer has been shown in a French cohort of adults (n = 31) suffering from NPD-A and -B, in which five patients presented with cancer, two of whom had lung cancer, and two patients presented MGUS (Table 2) [47]. In addition, cancer has been reported as the cause of death in 9% of patients with NPD-B [48].

To a lesser extent, metachromatic leukodystrophy (MLD) could be associated with gallbladder carcinoma. MLD is caused by the deficient activity of arylsulfatase A, encoded by the *ARSA* gene, which results in the accumulation of sulfatide and lysosulfatide (Table 1). Three clinical subtypes are described: the late-infantile (0–2.5 years), juvenile (2.5–16 years), and adult (>16 years) forms [49]. Patients with this disease present a progressive neurological involvement with leukodystrophy at MRI; gallbladder complications such as hemobilia and cholecystitis are also described [50] (Table 2). Interestingly, in a cohort of 34 patients, an increase in the prevalence of gallbladder polyposis, a precancerous condition for gallbladder carcinoma, has been observed [51]. In addition, cases of gallbladder carcinoma with liver metastases have been reported in two patients with an adult form of MLD [51,52] and a case without metastases in one patient with a juvenile form [53]. It should be noted that in these patients, the carcinoma occurred earlier than in the general population (56–63 years).

Taken together, these observations suggest a potential link between the occurrence of cancer and the above four sphingolipid storage diseases. In some cases, a common location was observed between typically affected organs and associated cancer, e.g., cytopenia and blood cancers in GD or chronic kidney disease and kidney cancer in FD (Table 2). It seems that there is no correlation between the plasma concentration of lysoSLs and the risk of developing malignancies. The development of cancer seems to be associated with mild or late-onset forms of sphingolipid storage diseases. For instance, the majority of Gaucher patients having cancer present the classical mild form type 1. Indeed, a very slight augmentation of plasma GlcSph levels is associated with the occurrence of GlcSph-reactive IgGs [62]. Such IgGs have already been described in some GD patients with gammopathy and could be causally involved (see Section 3.5). Moreover, in the studied cohorts of Fabry patients with cancer, there are fewer males than females [44,45]. Females usually have lower lysoGb3 levels, present a milder, late-onset form of the disease, and very rarely develop kidney complications compared to males. No patient who is affected by a severe form of NPD and suffers from cancer has been reported [47,48]. These observations could be explained by life expectancy, which is longer in the milder forms, thus allowing cancer development. Moreover, the above findings were based on retrospective or cross-sectional studies, which limits the scientific evidence level. This would require further randomized studies in order to confirm the association between sphingolipid storage diseases and cancer and, if necessary, to determine risk factors such as splenectomy, treatment, SLs, and lysoSL levels.

Although the underlying mechanisms remain unknown, the accumulation of sphingolipids and their corresponding lysoSLs could be implicated in the increased risk of cancer. In GD, the therapeutic splenectomy may lead to the release of undegraded lipids and could be associated with the occurrence of multiple cancers in the same patient [36]. In the FGDR, three out of five patients with two distinct cancers were splenectomized (data were unavailable for the remaining two). A common treatment for Gaucher and Fabry patients is enzyme replacement therapy (ERT), which is based on the administration of exogenous recombinant enzymes. Considering that these treatments improve clinical symptoms and reduce circulating lysoSL levels, they would be expected to be protective against cancer development, which has already been reported [63]. However, reports in the literature are inconsistent in this regard [64], which may suggest that the build-up of these lipids could result in early irreversible protumor events.

In the following chapter, the potential protumor effects of lysoSLs are discussed. The possible role of their parental lipids (GlcCer, Gb3, and SM) in cancer has already been discussed elsewhere [30,65,66,67].

## 3. Can Lysosphingolipids Be Related to Cancer Features?

Here, we attempt to review the potential protumor roles of lysoSLs according to the hallmarks of cancer published by Hanahan and Weinberg in 2000 and a recently completed study [68]. LysoSLs could activate several signaling pathways in tumor cells and/or tumor stromal cells, allowing cancer cells to proliferate, survive, and invade other tissues.

### 3.1. Lysosphingolipids and ‘Sustained Proliferation and/or Resistance to Cell Death’

The first report on the mitogenic properties of a lysoSL described the ability of lysoSMs (i.e., sphingosylphosphocholine or sphingosylphosphorylcholine) to stimulate the DNA synthesis and cell division of several untransformed and transformed cells [69]. Since then, numerous studies have shown that lysoSM controls cell proliferation, either positively or negatively, depending on cell type. However, whereas lysoSM treatment stimulates the proliferation of several normal cells, it would rather have an antiproliferative effect on cancer cells [70,71]. This latter phenomenon is associated with rapid increases in cytosolic free Ca2+ and the phosphorylation of the focal adhesion kinase p125FAK in ovarian cancer cell lines [72]. It is mediated by inhibition of both the PI3K-Akt and the mitogen-activated protein kinase (MAPK) signaling pathways in thyroid FRO cancer cells [73] or the inhibition of the G1/S transition of the cell cycle in numerous human pancreatic cancer cells [74]. Moreover, lysoSM strongly inhibits the proliferation of MDA-MB-435S breast cancer cells by regulating the Hippo-signaling pathway, causing an up-regulation of the tumor suppressive kinase Lats2 via S1P receptor 2 (S1P2) and Akt inhibition [75]. Besides its antiproliferative effects, lysoSM is able to trigger both autophagy and apoptosis in triple-negative breast cancer MDA-MB-231 cells [76]. Autophagy and apoptosis are catabolic pathways interconnected by several molecular nodes of crosstalk. Autophagy acts as a double-edged sword in cancer, preventing cancer development through the degradation of oncogenic molecules or by inducing the survival of tumor cells in hypoxic or low-nutrition environments. Moreover, autophagy plays a pivotal role in some cancer hallmarks, including the epithelial–mesenchymal transition (EMT) process and cancer stem cell promotion. Quite surprisingly, lysoSM-induced autophagy, which is mediated by Akt/p38 MAPK signaling, was found to antagonize apoptosis through the inhibition of JNK signaling [76]. Similar results have been reported in non-small-cell lung cancer A549 and H157 cells, in which lysoSM treatment stimulates autophagy through Akt/mTOR and p53 signaling pathways and inhibits apoptosis at the same concentrations [77]. Moreover, the evasion of apoptosis encourages tumor progression, in part through the emergence of resistant variants with great metastatic potential [78].

Lysoglycosphingolipids (lysoGSLs), in particular, GlcSph and GalSph, have also been reported to affect cell growth. Of note, most studies that have examined the effects of these lipids were conducted on cultured non-cancer cells, using cell types known to be affected in sphingolipid storage disorders. For instance, GlcSph was shown to reduce the cell proliferation of human bone cells (osteoblasts, mesenchymal stem cells, and myeloma plasma cells) and the differentiation of peripheral blood mononuclear cells (PBMCs) into osteoclasts [79,80]. In neuronal cells, GlcSph reduces cell growth, promotes apoptosis, and activates mTORC1, a potent inhibitor of autophagy [81]. This autophagy defect could contribute to cell death, as observed in GD [82]. Likewise, GalSph decreases the cell growth of human and murine oligodendroglial cell lines (MO3.1 and OLP-II) and astrocytes [83,84,85,86]. Several pathogenic signaling pathways [83,84] involving either S1P receptors (S1PRs) [87] or cPLA2 [88] have been described. Finally, lysoGM1a and lysoGM2 can induce the apoptosis of murine neuroblastoma Neuro2a cells [89].

Hence, whereas lysoSLs generally elicit antiproliferative effects on many cell types, including tumor cells in vitro, they can differentially affect autophagy, leading to the inhibition or activation of apoptosis and cell death, depending on the lysoSL implicated.

### 3.2. Lysosphingolipids and “Activating Invasion and Metastasis”

Metastasis is the most life-threatening event in patients with cancer. During the metastatic cascade, loss of cell–cell adhesion capacity and modifications in cell–matrix interaction allow cancer cells to invade local tissue. These events are based on changes in the expression and function of proteins involved in the control of motility and the secretion of molecules to degrade the extracellular matrix (ECM).

LysoSM favors the migration or invasion of many cancer cells, including pancreatic, breast, and lung cancer cells [71]. LysoSM-induced cell migration is associated with increased elastic properties in PANC-1 pancreatic cancer cells [90] and the perinuclear reorganization of keratin-8 intermediate filaments in PANC-1 pancreatic cancer cells and A549 lung cancer cells [91]. Depending on cell type, lysoSM-induced keratin-8 phosphorylation and reorganization were shown to be mediated by the G protein-coupled orphan receptor GPR12 [92], the small GTPase RhebL1 [91], or transglutaminase-2, a Ca^2+^-dependent enzyme [93]. Exogenous lysoSM can also induce cell invasion by stimulating the expression and secretion of the ECM-degrading enzyme MMP3 in an ERK-dependent manner in several breast cancer cells [94]. Moreover, lysoSM can induce EMT in lung [95] and breast [96] cancer cells by reducing E-cadherin expression and, inversely, enhancing the expression of N-cadherin and vimentin. LysoSM also triggers TGF-β1 expression/secretion and the Wnt-signaling pathway; all these events mediate cancer cell migration and invasion in vitro as well as metastasis in a mouse model of lung metastasis [95]. Interestingly, lysoSM can stimulate the release of connective tissue growth factor (CTGF) [97], which is known to induce tumor cell EMT and facilitate tumor growth and metastasis by promoting the deposition and orientation of collagen I fibers at the primary tumor stroma [98].

Although the promigratory effects of lysoSM have been well documented, anti-migratory effects have also been reported. LysoSM shares a similar structure to S1P, and several cellular effects of lysoSM can be explained by low-affinity binding to and activation of S1PRs [99]. The dual effect of lysoSM on migration may depend on the expression pattern of S1PR subtypes. For instance, lysoSM and S1P inhibit the migration and invasion of B16 murine melanoma cells via S1P2 [100]. It is tempting to speculate that lysoSM promotes the migration/invasion of cancer cells through binding to S1P1 or S1P3, both of which exert promigratory effects; however, this hypothesis remains to be demonstrated [100]. Of interest is the finding that lysoSM concentration is markedly elevated in ascites samples from ovarian cancer patients compared to patients with non-malignant diseases, suggesting a potential role as an extracellular signaling molecule in tumor progression [101]. It was proposed that lysoSM could exert biological actions not shared with S1P; however, no high-affinity lysoSM-specific membrane receptor has yet been conclusively identified. Indeed, after the characterization of an orphan receptor named ovarian cancer G-protein-coupled receptor 1 (ORG1) and other receptors with sequence homology, such as high-affinity lysoSM receptors, subsequent studies have failed to reproduce important features of the original reports, and those were finally retracted.

Contrary to lysoSM, lysoGSLs would hamper cell migration. Indeed, GlcSph slows human umbilical vein endothelial cells (HUVEC) wound healing closure [102], and lysosulfatide inhibits the migration and adhesion of B35 neuroblastoma cells through calcium release, leading to cell retraction and rounding [103].

### 3.3. Lysosphingolipids and “Unlocking Phenotypic Plasticity”

Unlocking phenotypic plasticity, which confers to cancer cells the ability to escape from the state of terminal differentiation, is a critical component of tumor aggressiveness in multiple cancer types [68]. This plasticity is related to the disruption of cellular differentiation (e.g., dedifferentiation from mature to progenitor states), blocked differentiation from progenitor cell states, and transdifferentiation into different cell lineages. In melanoma, loss of microphthalmia-associated transcription factor (MITF), the master regulator of melanocyte differentiation, is associated with the reactivation of neural crest progenitor genes and leads to invasive tumors. Interestingly, lysoSM is able to reduce the expression of MITF and its downstream target tyrosinase and, therefore, to induce hypopigmentation in human melanocytes via the activation of ERK [104]. However, whether the lysoSM-induced loss of MITF is associated with the reactivation of neural crest progenitor genes and melanoma progression is unknown. In contrast, there is some evidence that lysoSM stimulates the differentiation of promyelocytic leukemia cells. As a matter of fact, treatment of NB4 cells with lysoSM results in a marked increase in cell differentiation compared to the untreated controls, as demonstrated by the increased expression of the β2 integrin CD11c/CD18. This effect is also mediated by the ERK cascade but is independent of S1P1 because NB4 cells do not express this receptor [105].

So far, there is no report on lysoGSLs and phenotypic plasticity.

### 3.4. Lysosphingolipids and “Inducing or Accessing Vasculature”

Angiogenesis is a key feature of tumor progression and metastasis as the formation of new blood vessels from pre-existing vessels provides the principal route by which tumor cells enter the circulation. Although the following effects have never been described in a tumor context, lysoSM appears to be a critical regulator of angiogenesis by inducing capillary-like tube formation via vascular endothelial growth factor (VEGF) receptor 2 transactivation and the activation of both PI3K and Akt [106]. LysoSM also induces the chemotactic migration of human and bovine endothelial cells in a manner similar to that exerted by VEGF [107]. The molecular mechanisms of lysoSM-induced angiogenesis involve an increased expression of the urokinase-type plasminogen activator (uPA), which is known to promote pro-angiogenic signaling through the proteolytic degradation of ECM and upon binding to uPAR/CD87 receptors [108]. Moreover, lysoSM dose-dependently increases the activity of ecto-5′-nucleotidase (CD73) in HUVEC [109]. This effect could lead to adenosine production, an immunosuppressive metabolite that also exerts mitogenic effects on vascular cells and may contribute to angiogenesis [110].

So far, there have been no reports on lysoGSLs and angiogenesis.

### 3.5. Lysosphingolipids and “Tumor Promoting Inflammation/Avoiding Immune Destruction”

Smoldering inflammation favors cancer progression as it contributes to the proliferation and survival of tumor cells, angiogenesis, metastasis, and the subversion of adaptive immunity. LysoSM could exert a pro-inflammatory role, thanks to its capacity to recruit immune cells such as natural-killer-cell-producing cytokines [111] and to trigger the release of lipid mediators and cytokines, including leukotriene B4 [112], interleukin-6 (IL-6) [113,114], and tumor necrosis factor-α (TNF-α) [115] in normal cells. TNF-α production was also reported upon repeated exposure of astrocytes to lysoSM, mimicking the pathological conditions observed in NPD-A [116]. Mechanistically, as reported in cerebral artery vascular smooth muscle cells, lysoSM can stimulate pro-inflammatory transcription factors, nuclear factor-κB (NF-κB), and CCAAT-enhancer-binding protein in a p38MAPK-dependent manner [117], leading to the release of cytokines and monocyte chemoattractant protein-1 (MCP-1) [117]. Similarly, lysoSM also activated NF-κB and AP-1 via p38MAPK and Jak/STAT3 to produce MCP-1 in HUVEC [118].

Interestingly, lysoSM induces IL-8 expression and secretion in ovarian cancer cells [119], probably creating a tumor microenvironment that could be conducive to the recruitment of neutrophil populations. Tumor-associated neutrophils have immunosuppressive functions, stimulate angiogenesis, and favor tumor progression [120]. However, the immunosuppressive properties of lysoSM may be counterbalanced by its ability to reduce IL-1β-stimulated prostaglandin E2 (PGE2) formation [121], PGE2 being an important player in tumor immune evasion. Moreover, the accumulation of lysoSM in peripheral tissues may favor a T helper type 1-mediated immune response through the release of IL-12 and IL18 from dendritic cells [122]. However, the relevance of these in vitro data, obtained with a high concentration (40 µM) of lysoSM, remains questionable in the context of NPD.

Increasing evidence also implicates lysoGb3 as a mediator of inflammation. Indeed, lysoGb3 triggers Notch1 signaling and the subsequent activation of the NF-κB pathway, resulting in the production of pro-inflammatory cytokines in human podocytes [123]. Accordingly, systemic injection of lysoGb3 into mice resulted in the whole kidney up-regulation of Notch ligand Jagged1 and chemokine (MCP1, RANTES) expression, leading to increased numbers of macrophages in glomeruli. The shear stress induced by the lysoGb3-mediated remodeling of the arterial wall may increase the inflammatory potential of endothelial cells [124]. Interestingly, it was also reported that cultured peripheral blood mononuclear cells from Fabry patients exhibit a higher pro-inflammatory cytokine expression profile than that of healthy individuals [125]. Moreover, reduced proportions of blood monocytes, CD8+ T-cells, and dendritic cells have been reported in Fabry patients, together with increased percentages of total lymphocytes and B-cells [126]. Whether these changes are solely due to lysoGb3 remains to be established.

GlcSph is another lysoSL that affects innate and adaptive immune responses. The first evidence of GlcSph is a marked elevation of chitotriosidase, a macrophage activation marker, as well as an increased concentration of pro-inflammatory cytokines such as IL-6 and TNFα in the plasma of Gaucher patients [36,127,128,129,130,131]. In addition, the long-term subcutaneous administration of high doses of GlcSph in C57BL/6JRj mice replicated the visceral inflammatory response observed in patients [132,133,134]. Moreover, the activation of B-cell proliferation could result from the production of IL-6 [127] but also from the activation of type II Natural Killer (NK) T-cells against GlcCer or GlcSph [135]. These results highlight the potential role of GlcSph in B-cell lymphoma development. Although the specific implication of GlcSph was not demonstrated, sporadic B lymphoma was seen in aged mice affected with GD [136].

The monoclonal immunoglobulin (IgG) found in Gaucher patients and in a conditional Gaucher mouse model reacts against GlcSph [137]. The IgG level decreased following pharmacological inhibition of GlcCer synthase, an intervention that reduces the production of both GlcCer and GlcSph. Quite surprisingly, GlcSph-reactive IgGs were also detected in patients suffering from sporadic monoclonal gammopathy [137] and myeloproliferative neoplasms [62]. These results indicate that GlcSph could have an immunogenic role, linking auto-immunity and inflammation [138].

To elucidate the specific role of GlcSph in inflammation (and neuroinflammation in particular), experiments were conducted using a Gaucher zebrafish model where the expression of acid ceramidase, the enzyme responsible for GlcSph production, was ablated. This manipulation caused no reduction in the expression of inflammatory cytokines such as IL-1β and TNFβ in the brain [139], suggesting the minor contribution of GlcSph to the neuroinflammation in this model.

Similar to GlcSph, lysosulfatide has been recognized as a potent ligand for murine type II CD1d-restricted NKT cells [140], indicating a potential involvement in tumor immunity.

In sum, the data reviewed in this section suggest that lysoSM behaves as a pro-tumorigenic metabolite. This effect could be linked to its structure, which is similar to that of S1P, a well-known lysolipid that has potent protumorigenic activities. The effects of lysoSM vary according to cell type and extracellular concentration. They may also depend on the nature of the S1PR that mediates these effects, the potential competition with S1P, or the autotaxin-mediated transformation of lysoSM into S1P. These effects will ultimately depend on other microenvironmental stimuli.

Moreover, a number of the aforementioned results should be taken with caution because most of the in vitro experiments were conducted on different cell types. Indeed, one could conclude that lysoGSLs have anti-tumorigenic properties, but these observations were made on non-cancer cells. Moreover, the choice of concentrations (up to 10 µM) of the tested lysoSLs is often questionable. Physiological levels of these lipids in plasma approximate the nanomolar range; in a pathological context such as sphingolipid storage disorders, their plasma concentrations can rise by one or two orders of magnitude [13,27]. Although very little information on their concentrations in organs is available, it is likely that the concentrations can reach the submicromolar range.

In the first two chapters, a likely association between sphingolipidoses and cancer occurrence has been emphasized. Thus, it is tempting to speculate that the accumulation of these lysoSLs in patients could facilitate tumor formation and/or progression. Faced with a lack of knowledge about lysoGSLs in cancer, we undertook a study to examine the role of one lysoGSL, namely, GlcSph, in cancer.

## 4. What Is the Role of Lysosphingolipids in Sphingolipid-Storage-Disorder-Associated Cancers? The Example of Glucosylsphingosine in Melanoma

Evidence has accumulated for a dysregulation of sphingolipid metabolism in numerous cancers, including cutaneous melanoma [141]. Moreover, melanoma has been reported in Gaucher patients (see Table 2). In this last part, we investigate the role of the lysoSL GlcSph in melanoma progression.

### 4.1. Melanoma Growth Is Promoted in a Mouse Model of Gaucher Disease

First, we studied melanoma growth in a well-established, type 1 GD mouse model carrying the homozygous point mutation p.D409V (Gba1^D409V/D409V^) [142,143]. Murine melanoma cells (B16F10) were injected subcutaneously in these mutant mice and in their control, age-matched heterozygous littermates (Gba1^D409V/wt^). Tumor growth (Figure 2A) and tumor weight (Figure 2B) were significantly increased in Gaucher mice compared with their control counterparts. As expected, the enzyme activity of GCase in the liver and spleen was significantly decreased in Gaucher mice (Figure 2C). In addition, the concentrations of GlcCer and GlcSph in these two organs were significantly higher in GD mice than in control ones (Figure 2D,E). The quantification of ceramide, SM, dihydroceramide, GM3, and lactosylceramide levels did not show any significant differences between the groups [144]. These results on tumor growth are consistent with those found in a similar GD mouse model D409V/null [30]. Thus, melanoma growth seems to be facilitated in GD mice, which suggests that the Gaucher microenvironment, encompassing the lipids GlcCer and GlcSph, could promote tumor development. This finding is in line with previous observations showing that, similar to Gaucher patients, who are prone to hematological malignancies, a conditional mouse model of GD developed significantly more B-cell lymphomas than control animals [136]. Of note, the early administration of an inhibitor of UDP-GlcCer transferase resulted in a reduction of plasma GlcSph levels and protected against lymphoma and myeloma development [145], possibly suggesting that GlcSph was implicated in the higher susceptibility of Gaucher animals. To test this hypothesis, we investigated the potential role of GlcSph in melanoma progression.

### 4.2. GlcSph Dose-Dependently Decreases Human Melanoma Cell Growth In Vitro

To examine the role of GlcSph, A375 cells, a human metastatic melanoma cell line, were treated with various concentrations of GlcSph (up to 5 µM) or ethanol (vehicle) for 4 days. Cell growth was evaluated using the IncuCyte S3 time-lapse microscopy system, a live cell-imaging incubator, through the determination of cellular confluence. Figure 3A depicts a concentration-dependent reduction in cellular confluence during the first 24 h. These results were also observed in GlcSph-treated WM35 cells, a primary human melanoma cell line (Figure 3B). To further assess the role of GCase inhibition in cell growth, A375 cells were incubated with conduritol B epoxide (CBE), a GCase irreversible and specific inhibitor [147], resulting in a significant decrease in enzyme activity (Figure 3C). However, only a trend of decreased cell growth in CBE-treated cells compared to vehicle-treated cells was observed (Figure 3D). To go further, we analyzed SL levels in cells treated with CBE for 48 h or GlcSph for 24 h. No significant increase in the GlcSph level was observed after CBE treatment, whereas its concentration was very high in GlcSph-treated cells (Figure 3E). These changes in GlcSph concentrations could explain the above difference in cell growth.

Thus, exogenously added GlcSph and, possibly, the intracellular GlcSph produced upon the pharmacological inhibition of GCase decrease the proliferation of human melanoma cells in a concentration-dependent manner. A cytotoxic effect of GlcSph has already been reported in other cell types in a micromolar concentration range of GlcSph [80,148]. The concentrations we used are consistent with the GlcSph plasma concentration found in Gaucher patients [13,149]. The intraepidermal GlcSph concentrations in Gaucher patients are unknown, but in Gaucher mouse models (D409V; C* and N409S-/-), they are elevated [150]. This suggests that melanocytes could be surrounded by a local accumulation of GlcSph. Moreover, a significant increase in sphingosine (Sph) (Figure 3F) and a slight increase (*p* = 0.05) in sphingosine 1-phosphate (S1P) levels were observed only in GlcSph-treated cells, while the quantification of ceramide, sphinganine, GlcCer, SM, dihydroceramide, and GM3 levels did not show any differences between groups [144]. These results are consistent with the reported elevation of sphingosine and S1P levels in the plasma of Gaucher patients [149]. Pro-apoptotic effects of sphingosine have already been described in different cell types but not in melanoma cells [151]. Although S1P is considered an anti-apoptotic metabolite [152], it may lead to cell death by apoptosis, as shown in prostate cancer cells, possibly via its dephosphorylation to sphingosine [153]. Sphingosine comes from the N-deacylation of ceramide by ceramidases and can be phosphorylated by the sphingosine kinase to produce S1P (Figure 1). Here, the observed increase of sphingosine and S1P in GlcSph-treated cells could originate from GlcSph catabolism. The enzyme involved in this catabolism could be the lysosomal GCase or GBA2, a glucosylceramidase located in the ER. Indeed, the double gba1:gba2 knockout in zebrafish larvae led to a higher level of GlcSph than in single gba1 knockout larvae [154]. Lastly, GlcSph has also been reported to be able to alter the lipid composition of plasma membranes, as shown with GalSph, which shares a similar chemical structure and build-up to the neurodegenerative Krabbe disease. Indeed, enrichment of GalSph in the inner layer of the plasma membrane promotes neuronal death [155].

### 4.3. GlcSph Promotes Necrosis and Produces Multinucleated Melanoma Cells

To further characterize the reduction of cell growth upon GlcSph administration, cellular death was explored. A375 cells, treated or not with GlcSph for 24 h, were stained with Annexin V-FITC and 7-AAD and analyzed by flow cytometry. The percentage of necrotic cells was significantly higher in GlcSph-treated cells than in control cells, while no difference was observed in the proportion of apoptotic cells (Figure 4A). These findings indicate that GlcSph decreases melanoma cell growth in vitro by promoting necrosis. Whereas the underlying mechanisms remain unknown, it is interesting that the development of neurodegeneration in the neuronopathic forms of GD (i.e., types 2 and 3) would result from necroptosis, a programmed form of necrosis, which could be induced by GlcSph [156,157]. Moreover, the involvement of Sph and S1P has been observed in nutrient-deprivation-induced necroptosis in murine embryonic fibroblasts [158]. Interestingly, necroptosis has also been reported as a mechanism participating in tumor growth in pancreatic cancer through the release of immunosuppressive factors [159]. Although necroptosis of GlcSph was not documented in our model, it could explain how melanoma growth was exacerbated in GD mice (Figure 2), whereas the in vitro cell growth was reduced under GlcSph (Figure 3).

Concurrently, morphological analysis of A375 cells after exposure to GlcSph (5 µM) for 48 h showed larger and multinucleated cells, as illustrated in Figure 4B–E. To quantify the presence of multinucleated cells, the cell cycle was analyzed and DNA content measured. As depicted in Figure 4F–H, the percentage of cells with 8N and 16N DNA content, relative to total cell count, was augmented after GlcSph (5 µM) treatment. The percentage of polyploidy reached a maximum of 7% of cells with 8N after 48 h incubation (Figure 4G), demonstrating that GlcSph induced the formation of multinucleated cells in a human melanoma cell line. Of note, giant multinucleated cells have been described in histopathologic analyses of melanoma [160,161], but their presence does not seem to be associated with poor prognosis [162]. Usually, aneuploidy confers pro-tumoral properties to cancer cells [163]. Because of their genetic instability, multinucleated cells are usually programmed for cell death. However, the genetic rearrangement could confer to cancer cells selective advantages that favor tumor growth, migration, and resistance to treatment [163]. Our observations might suggest that the multinucleated cells represent a fraction of cells resistant to GlcSph-induced death.

The appearance of multinucleated cells elicited by GlcSph has already been observed in non-melanoma cell lines, including the human myelomonocytic U937 cell line [102,164] and with GalCer. GalSph was proposed to inhibit cytokinesis by impacting actin reorganization [164] or by disrupting SM clusters at the plasma membrane and suppressing the formation of phosphatidylinositol 4,5-bisphosphate at the cleavage furrow [152]. Whether GlcSph and GalSph induce polyploidy through the same molecular mechanisms remains to be determined.

## 5. Conclusions

This review aims to provide an overview of lysoSLs and their potential role in cancer. Although they represent a very small fraction of total lipids, they are considered bioactive lipids. Their biological effects likely depend on their lysoSL nature (with a distinction between lysoSMs and lysoGSLs) and on cell type. Current evidence indicates that these lipids come from the deacylation of their parental SL by aCDase. Their mechanisms of action are not fully understood. LysoSM can act on G-protein-coupled receptors (S1PRs) [92,99]. On the other hand, GalSph may alter the lipid composition of plasma membranes and, as a consequence, affect signaling pathways that depend on lipid rafts [165] and/or tyrosine kinase receptors [155].

As discussed, there are very little data on the effects of lysoGSLs in the context of cancer. However, these lysoSLs, nowadays considered useful biomarkers of sphingolipid storage diseases, could be involved in the occurrence of malignancies in patients suffering from these inherited metabolic disorders. LysoSLs may act directly on cancer cells to promote their growth and/or resistance to anticancer regimes or indirectly on the cellular tumor microenvironment by triggering protumorigenic (e.g., pro-inflammatory and immunosuppressive) responses. For example, GlcSph could be a good candidate in the genesis of lymphoma and myeloma in patients with GD. Regarding cutaneous melanoma, we have shown that the treatment of melanoma cells results in cell necrosis and the build-up of multinucleated melanoma cells. Such modifications in cell cycle and death programs could play a significant role in the progression of skin melanoma. Indeed, as previously mentioned, necroptosis could induce immunosuppression, and the multinucleated cells may represent a way to resist cell death, thereby leading to tumor growth.

Another hypothesis linking sphingolipids, GD, and cancer implicates α-synuclein. As a matter of fact, it is now established that carrying a *GBA* pathogenic variant gene represents the greatest risk factor for developing Parkinson’s disease (PD) [166]. PD is a neurodegenerative disease in which α-synuclein aggregates are part of the pathophysiological process [167,168]. The accumulation of GlcCer and GlcSph that accompanies GCase deficiency could be implicated in this protein aggregation [169,170]. Interestingly, patients affected with PD have a high risk of developing cutaneous melanoma [171,172]. Whereas α-synuclein aggregation induces neuronal cell death, some observations account for its capacity to promote some cancers, such as melanoma. For example, α-synuclein has been reported to favor cell survival both in in vitro and in vivo melanoma models [173,174], raising the possibility of a link between GlcCer/GlcSph, α-synuclein aggregation, and melanoma.

Taken together, current knowledge points to the potential role of lysoSLs in the occurrence of cancer. Further investigation of these metabolites in relation to cancer should be supported. This certainly includes systematic studies on their biological effects and their modes of action. Also needed are accurate measurements by mass spectrometry of their concentrations in tumor samples and plasma at the time of diagnosis and during the follow-up of patients with cancer. LysoSLs are usually quantified in the plasma of patients suspected of suffering from sphingolipidoses, but plasma may not be the appropriate milieu to reveal their possible role in cancer development. Indeed, malignancies seem to occur in patients affected with mild forms of sphingolipid storage diseases, i.e., under conditions where the plasma levels of lysoSLs are lower than in patients with infantile severe forms, and a long-term cumulative effect operates. Finally, future work should determine whether the incidence of cancer can be reduced in patients who receive long-term enzyme replacement (ERT) or substrate reduction (SRT) therapy, as these treatments lead to a considerably reduced production of lysoSLs.

## Figures and Tables

**Figure 1 cancers-14-04858-f001:**
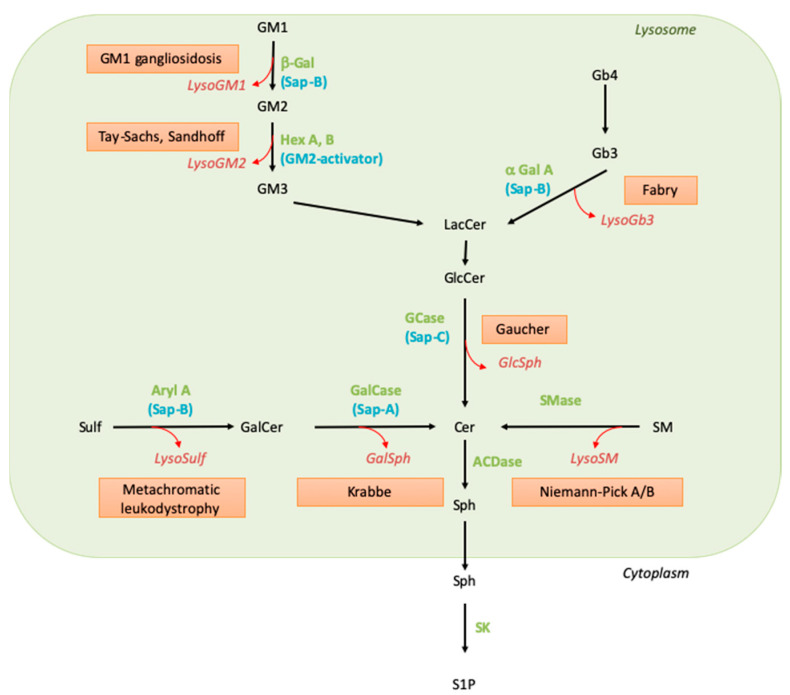
Sphingolipid catabolism and associated diseases. Green names correspond to the enzyme names, blue names to their activators, the red boxed texts contain the disease name, and the red and italic names indicate the corresponding lysoSL. Abbreviations: α-Gal: alpha-galactosidase; ACDase, acid ceramidase; Aryl A, arylsulfatase A; β-Gal, beta-galactosidase; Cer, ceramide; GalCase, galactosylceramidase; GalSph, galactosylsphingosine; GCase, glucosylceramidase; GlcCer, glucosylceramide; GlcSph, glucosylsphingosine; Hex, hexosaminidase; LacCer, lactosylceramide; LysoSulf, lysosulfatide; Sap, saposin; SK, sphingosine kinase; SM, sphingomyelin; SMase, sphingomyelinase; Sph, sphingosine; Sulf, sulfatide; S1P, sphingosine 1-phosphate.

**Figure 2 cancers-14-04858-f002:**
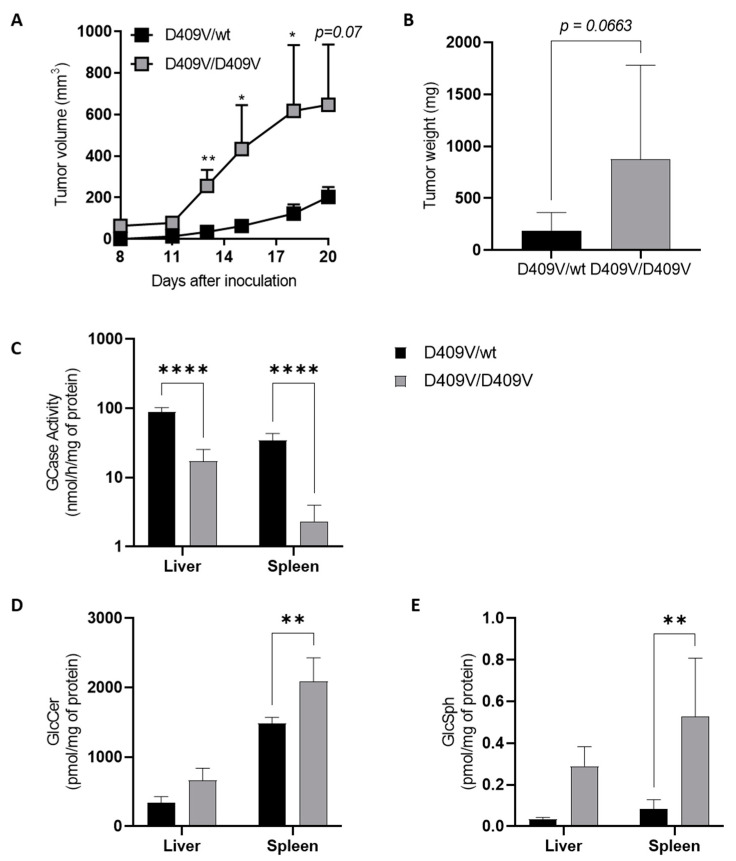
Melanoma tumor growth is increased in a Gba1^D409V/D409V^ Gaucher mouse model. Murine B16F10 melanoma cells (3 × 10^5^) were injected subcutaneously into Gba1^D409V/D409V^ (D409V/D409V) female mice and heterozygous (D409V/wt) littermates having the same mixed genetic background (19–22 weeks of age). (**A**) Tumor volumes were measured every 3 days. (**B**) Tumor weights were measured 25 days after tumor inoculation. Data are expressed as means +/− SEM of at least two independent experiments (n = 7–21 mice). (**C**) GCase enzyme activity was determined in lysates of liver and spleen isolated from D409V/D409V or heterozygous D409V/wt mice. Assays were performed in duplicate on samples of three to six animals. GlcCer (**D**) and GlcSph (**E**) were extracted by the Bligh and Dyer method [146] from the liver and spleen and analyzed by LC-MS (liquid chromatography coupled by mass spectrometry); assays were performed in duplicate on samples taken at day 20 of three to six animals (18–22 weeks of age). Statistical significance was determined by a Mann–Whitney test. *, *p* < 0.05; **, *p* < 0.01; ****, *p* < 0.0001.

**Figure 3 cancers-14-04858-f003:**
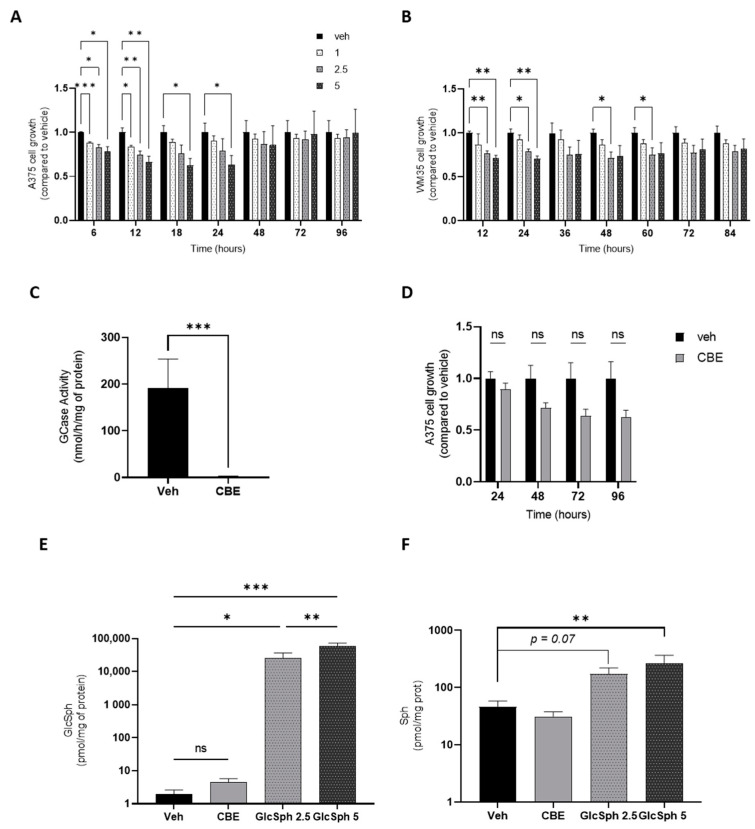
GlcSph decreases the growth of cultured human melanoma cells in a dose-dependent manner. (**A**,**B**) A375 and WM35 human melanoma cells were seeded into a 96-well plate (10,000 cells/well) and, 24 h later, treated either with different concentrations of GlcSph (1, 2.5, or 5 µM) or with ethanol (veh). Cell growth was assessed by measuring the cellular confluence using an IncuCyte S3 time-lapse microscopy system. For each sample, the cellular confluence was measured in duplicate at different incubation times for 4 days. (**C**) GCase enzyme activity in A375 cells treated with CBE (30 µM) or DMSO (veh) for 48 h. (**D**) A375 cells were seeded into a 96-well plate and, after 24 h, treated either with CBE (30 µM) or DMSO (veh). Cell growth was evaluated as described in (**A**,**B**). GlcSph (**E**) and Sph (**F**) were extracted as described in [13], and concentrations were determined by liquid chromatography-tandem mass spectrometry in CBE- or GlcSph-treated cells compared to controls. For all experiments, data are expressed as means +/− SEM of three independent experiments. Statistical significance was determined by a two-way mixed ANOVA with Geisser–Greenhouse correction (**A**,**B**,**D**) or a Mann–Whitney test (**C**,**E**,**F**). *, *p* < 0.05; **, *p* < 0.01; ***, *p* < 0.001; ns, not significant compared to vehicle-treated cells.

**Figure 4 cancers-14-04858-f004:**
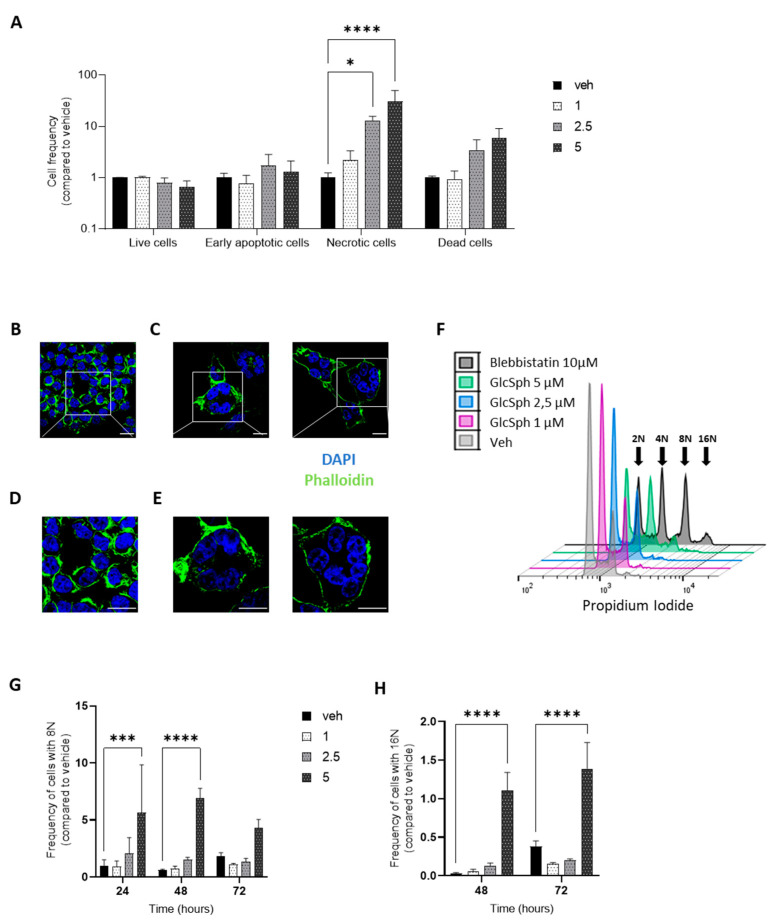
GlcSph induces necrosis and produces multinucleated cells in A375 human melanoma cells. (**A**) A375 cells were treated for 24 h with either different concentrations of GlcSph (1, 2.5, and 5 µM) or with ethanol (veh), stained with Annexin V-FITC and 7-AAD, and analyzed by flow cytometry; 20,000 events were recorded and sorted as follows: live cells (Annexin V-negative/7-AAD-negative), dead cells (Annexin V-positive/7-AAD-positive), early apoptosis (Annexin V-positive/7-AAD-negative), and necrosis (Annexin V-negative/7-AAD-positive). Histograms represent the cell frequency of GlcSph-treated cells compared to vehicle-treated cells. (**B**–**E**) A375 cells were treated for 48 h with ethanol (**B**,**D**) or GlcSph 5 µM (C and E), fixed and stained with phalloidin (for F-actin, green), and nuclei were counterstained with DAPI (for DNA, blue). Then, they were imaged by confocal microscopy (Zeiss) at 63× (**B**,**C**), with an additional upper zoom (×2) of the selected regions (white square) (**D**,**E**). Scale bar = 20 µm. (**F**–**H**) A375 cells were incubated either with different concentrations of GlcSph (1, 2.5, or 5 µM) or with ethanol (veh) for 24, 48, and 72 h. Then, cells were stained with propidium iodide (PI) and analyzed by flow cytometry. Figure (**F**) is a representative plot of cell cycle analysis of GlcSph- and vehicle-treated A375 cells for 48 h. To identify the peaks corresponding to abnormal ploidies, A375 cells were treated with blebbistatin (10 µM), a myosin II inhibitor (see arrows). The proportion of cells with 8N (**G**) and 16N (**H**) in GlcSph-treated cells compared to controls was quantified. For experiments in A, G, and H, data are expressed as means +/− SEM of three independent experiments. Statistical significance was determined by a two-way mixed ANOVA. *, *p* < 0.05; ***, *p* < 0.001; ****, *p* < 0.0001 compared to vehicle-treated cells.

**Table 1 cancers-14-04858-t001:** Primary accumulated SLs and lysoSLs in sphingolipid storage diseases. Of note, genetic conditions in which the accumulation of some SLs occurs as a secondary phenomenon are not listed here. The plasma concentrations of lysosphingolipids are indicated in brackets (control values).

Sphingolipid Storage Disease	Affected Gene and Protein	Primary Stored Sphingolipid	Major Accumulated Lysosphingolipid	Lysosphingolipid Plasma Concentration (nmol/L)	Ref.
GM1-gangliosidosis	*GLB1*, GM1 β-galactosidase	GM1-ganglioside	LysoGM1	0–40 (<0.07)	[12,13]
GM2-gangliosidosis type B (Tay-Sachs disease)	*HEXA*, α subunit of β-hexosaminidase A	GM2-ganglioside	LysoGM2	0–14.5 (ND)	[13,14]
GM2-gangliosidosis type 0 (Sandhoff disease)	*HEXB*, β subunit of hexosaminidase A	GM2-ganglioside	LysoGM2	0–118 (ND)	[12,13]
GM2-gangliosidosis type AB	*GM2A*, GM2 activator protein	GM2-ganglioside	LysoGM2 *	NA	[15]
Gaucher disease	*GBA*, acid β-glucosidase (β-glucosylceramidase)	Glucosylceramide	Glucosylsphingosine	46–427 (<3.5)	[6,13,16]
Gaucher disease, saposin C deficiency	*PSAP,* prosaposin	Glucosylceramide	Glucosylsphingosine	110 (<3)	[17]
Fabry disease	*GLA*, α-galactosidase A	Gb3 (trihexosylceramide)	LysoGb3 (globotriaosylsphingosine)	0.5–150 (<0.6)	[13,18]
Metachromatic leukodystrophy	*ARSA*, arylsulfatase A	Sulfatide	Lysosulfatide	ND	[19,20,21]
Metachromatic leukodystrophy, saposin B deficiency	*PSAP,* prosaposin	Sulfatide	Lysosulfatide *		[22]
Krabbe disease (globoid cell leukodystrophy)	*GALC*, β-galactosylceramidase	Galactosylceramide	Galactosylsphingosine (psychosine)	1.5–54 (<2)	[23,24]
Krabbe disease, saposin A deficiency	*PSAP,* prosaposin	Galactosylceramide	Galactosylsphingosine (psychosine)	12 (<3) #	[25]
Prosaposin deficiency	*PSAP,* proposin	Multiple SLs	Glucosylsphingosine, lysoGb3, lysoSM inconstantly	GlSph: 53–61 (<3) LysoGb3: 5–8 (<0.6) LysoSM: 15–22 (<15) GalSph: 0.9–1.7 (<1)	[26]
Niemann–Pick disease (types A and B)	*SMPD1*, acid sphingomyelinase	Sphingomyelin	LysoSM (sphingosylphosphocholine) and PPCS (“lysoSM509”)	lysoSM: 8–70 (<2.6) LysoSM509: 127–364 (<9)	[26,27]

Abbreviations: LysoSM, lysosphingomyelin; PPCS, N-palmitoyl-O-phosphocholineserine. NA: not available. ND: not detectable. The asterisk (*) denotes that accumulation of the lysoSL is likely but, to our knowledge, not yet demonstrated in patients. The symbol # indicates that the concentration was determined on dried blood spots.

**Table 2 cancers-14-04858-t002:** Overview of malignancies described in patients affected with Gaucher disease, Fabry disease, Niemann–Pick B disease, or metachromatic leukodystrophy.

Sphingolipid Storage Disease	Clinical Presentation/Affected Organs		Benign and Precancerous Lesions	Malignant Tumors
	Organ system	Sign/symptom(s)		
**Gaucher disease** **OMIM****#230800** [54]	Blood	Anemia, thrombopenia	Dysgammaglobulinemia-MGUS [29,34,55]	Hematological malignancies: [29,31,32,33] - Multiple myeloma - Lymphoma
	Viscera	Hepatosplenomegaly, gallstones		Digestive cancers - Hepatocellular carcinoma [29,31,32,35,56,57] - Colon carcinoma [31,32,57]
	Bone	Bone pain, bone infarcts, avascular necrosis, pseudo-osteomyelitis		
	Lung	Interstitial disease, fibrosis		Lung cancer [31,58,59]
	Skin	Collodion baby		Skin cancer: - Melanoma [31,32,57] - Basal cell carcinoma [31,32]
				Thyroid cancer [31,36,57]
**Fabry disease** **OMIM #301500** [60]	Nervous system	Periodic crises of acroparesthesia, sweating abnormalities	Meningioma [41,44]	
	Cerebrovascular disease	Stroke, transient ischemic attack		
	Eyes	Cornea verticillata		
	Skin	Angiokeratoma		Melanoma [44,45]
	Kidney	Nephropathy to end-stage renal disease		Renal cell carcinoma [37,39,40,44,45]
	Heart	Cardiac damage: left ventricular hypertrophy, cardiomyopathy, arrhythmia		
	Gastrointestinal tract	Nausea, vomiting, diarrhea	Colon polyp [44]	Colon cancer [42]
	Blood		MGUS [44]	Blood cancers [38,43]
**Niemann–Pick B** **OMIM #607616**	Viscera	Progressive hepatosplenomegaly, deterioration in liver function		Liver cancer [48]
	Blood	Thrombocytopenia	MGUS [47]	Multiple myeloma [48,61]
	Lung	Interstitial disease		Lung cancer [47]
	Bone	Osteopenia		Chondrosarcoma [48]
				Bladder cancer [47] Breast cancer [47] Thyroid cancer [47]
**Metachromatic leukodystrophy** **OMIM #250100**	Brain	Progressive neurological damage with leukodystrophy		
	Gallbladder	Hemobilia	Gallbladder polyposis [50]	Gallbladder carcinoma [51,52,53]

**Table 3 cancers-14-04858-t003:** Cancer prevalence in the French Gaucher Disease Registry and French population.

	French Gaucher Disease Registry (n = 445 Living Patients)	French Population, All Ages, Both Genders (International Agency for Research on Cancer, www.gco.iarc.fr, Accessed on 31 March 2021) (n = 65,273,512 Living People)	Odds Ratio, 95%CI, *p*-Value (Fisher’s Exact Test)
Cancer Type	Living Patients Number	Living Patients Number	
**All Cancers**	18	1,501,881	1.8 [1.03–2.81], *p* < 0.05
**Blood cancers** (**all**)	6	104,838	8.4 [3.1–18.4], *p* < 0.0001
- **Multiple myeloma**	3	18,442	23.8 [4.9–70.2], *p* < 0.0001
- **Non-Hodgkin lymphoma**	2	44,809	6.5 [0.8–23.8], *p* < 0.05
- **Essential thrombocytemia**	1	NA	NC
**Digestive Cancer** - **Cholangiocarcinoma and colon cancer**	2	86,328	3.4 [0.4–12.3], *p* = 0.11
**Lung Cancer**	2	59,708	4.9 [0.6–17.9], *p* = 0.06
**Skin Cancer** - **Non-melanoma cancers**	3 3	316,830 260,694	1.4 [0.3–4.1], *p* < 0.5
**Thyroid Cancer**	2	50,301	5.8 [0.7–21.2], *p* < 0.05
**Gynaecological Cancer** - **Breast and ovary cancers**	3	251,161	1.8 [0.4–5.2], *p* = 0.25
**Urological cancer** - **Prostate and bladder cancers**	2	312,121	0.9 [0.1–3.4], *p* = 1

NA: not available; NC: not calculated.

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
