# Peer review of "Potential Role of Sphingolipidoses-Associated Lysosphingolipids in Cancer"

_cancers, 2022, doi:10.3390/cancers14194858_

Round 1
Reviewer 1 Report
This review manuscript describes that patients with sphingolipid storage diseases categorized as lysosomal disease are susceptible to cancer based on epidemiological data in French. They also summarize reports concerning to the mechanism underlying of its susceptibility. The second half of this manuscript is their original research to investigate the effect of lysosphingolipid on cancer cell proliferation and integrity.
I think the review part is informative and worthwhile to publish. However, I cannot evaluate the second half of the research, because, this is review article.
In general, original research are published via sound evaluation process, including request of additional experiments, which make their conclusions to be more robust one. It also needs to include method section so that it can be reproduced. I recommend to omit their original research part if they want to publish this manuscript as review article.
I make some comments only in the review part.
l Concentrations of lysoSL in blood or organs in healthy subjects and those of lysosome disease patients are important information. It will be great help for reads who are interested in lysoSLs.
l Table 3 French population cell
n=65273512 living patientà65273512 people?
Author Response
Responses :
- As agreed with the Editors, we prepared a manuscript that will include a mini-review and original, unpublished experimental data. As a matter of fact, upon invitation by the Editors to submit an article to this Special Issue of Cancers, we wrote to the Editors as well as to Cancer Editorial Office (on Dec. 12, 2021) that our paper will include a review part and original results. This proposal was accepted by one of the Editors, Dr. Morjani, on Dec. 12, 2021 (cc. to Cancer Editorial Office).
We believe the experimental part of our manuscript provides a valuable illustration of the concepts that are discussed in the review part. Regarding the material and methods for this experimental part, methodological details are given in the legends to the figures.
In addition, we would like to point out that neither Reviewer#2 nor Reviewer#3 commented on this.
- As requested by the Reviewer, the plasma concentration of lysosphingolipids in control individuals and in patients with sphingolipidoses has now been included in Table 1.
- The text in Table 3 has been corrected (the word “patients” has been replaced by “people”).

Reviewer 2 Report
The paper is well written and shows interesting correlation between cancer and lysosphingolipids. However, despite the title describe a huge field of the sphingolipidoses and generation of different types of cancer, I think that the document context shows more the role description of the different lysosphingolipidoses in cancer diseases and a lower context focused on sphingolipidoses, perhaps due to the low prevalence of this type of alterations in patients with rare diseases. At the end, the document has a bias to Gaucher disease, that reflect a different approach to the expectation of the document title. I suggest to change or improve the document title.
Despite this, I consider it to be a very interesting and well-documented paper.
Author Response
Response : As recommended by this Reviewer, the title has been changed to “Potential role of sphingolipidoses-associated lysosphingolipids in cancer”.

Reviewer 3 Report
Please check the following if there are some words change required :
Table 2., Malignant tumors of Niemann-Pick B disease : thyroid or thyroid cancer.
Author Response
Response : The text in Table 2 has been corrected.

Round 2
Reviewer 1 Report
I recommend to publish your unpublished original data as not in review article but original paper. Your efforts are worthwhile to be evaluated by experts' criticism.